# Bioinspired Synthesis of Magnetic Nanoparticles Based on Iron Oxides Using Orange Waste and Their Application as Photo-Activated Antibacterial Agents

**DOI:** 10.3390/ijms24054770

**Published:** 2023-03-01

**Authors:** David Giancarlo García, Cristina Garzón-Romero, Mateo Alejandro Salazar, Karina J. Lagos, Kleber Orlando Campaña, Alexis Debut, Karla Vizuete, Miryan Rosita Rivera, Dario Niebieskikwiat, Maria J. Benitez, María Paulina Romero

**Affiliations:** 1Department of Materials, Escuela Politécnica Nacional (EPN), Quito 170525, Ecuador; 2Laboratorio de Investigación en Citogenética y Biomoléculas de Anfibios (LICBA), Centro de Investigación para la Salud en América Latina (CISeAL), Facultad de Ciencias Exactas y Naturales, Pontificia Universidad Católica del Ecuador (PUCE), Quito 170143, Ecuador; 3Centro de Nanociencia y Nanotecnología, Universidad de Las Fuerzas Armadas ESPE, Sangolquí 171103, Ecuador; 4Departamento de Física, Colegio de Ciencias e Ingenierías, Universidad San Francisco de Quito, Quito 170901, Ecuador; 5Departamento de Física, Escuela Politécnica Nacional (EPN), Quito 170525, Ecuador

**Keywords:** green synthesis, antibacterial PTT, iron oxide nanoparticles, orange peel extract, cytotoxicity, superparamagnetic behavior

## Abstract

Magnetic nanoparticles based on iron oxides (MNPs-Fe) have been proposed as photothermal agents (PTAs) within antibacterial photothermal therapy (PTT), aiming to counteract the vast health problem of multidrug-resistant bacterial infections. We present a quick and easy green synthesis (GS) to prepare MNPs-Fe harnessing waste. Orange peel extract (organic compounds) was used as a reducing, capping, and stabilizing agent in the GS, which employed microwave (MW) irradiation to reduce the synthesis time. The produced weight, physical–chemical features and magnetic features of the MNPs-Fe were studied. Moreover, their cytotoxicity was assessed in animal cell line ATCC RAW 264.7, as well as their antibacterial activity against *Staphylococcus aureus* and *Escherichia coli*. We found that the 50GS-MNPs-Fe sample (prepared by GS, with 50% v/v of NH_4_OH and 50% v/v of orange peel extract) had an excellent mass yield. Its particle size was ~50 nm with the presence of an organic coating (terpenes or aldehydes). We believe that this coating improved the cell viability in extended periods (8 days) of cell culture with concentrations lower than 250 µg·mL^−1^, with respect to the MNPs-Fe obtained by CO and single MW, but it did not influence the antibacterial effect. The bacteria inhibition was attributed to the plasmonic of 50GS-MNPs-Fe (photothermal effect) by irradiation with red light (630 nm, 65.5 mW·cm^−2^, 30 min). We highlight the superparamagnetism of the 50GS-MNPs-Fe over 60 K in a broader temperature range than the MNPs-Fe obtained by CO (160.09 K) and MW (211.1 K). Therefore, 50GS-MNPs-Fe could be excellent candidates as broad-spectrum PTAs in antibacterial PTT. Furthermore, they might be employed in magnetic hyperthermia, magnetic resonance imaging, oncological treatments, and so on.

## 1. Introduction

Bacterial infections are a potential cause for developing chronic diseases in patients, producing severe health problems and leading to death [1]. The uncontrolled use of antibiotics has contributed to the development of multidrug-resistant (MR) bacteria [2,3,4]. Since bacteria can proliferate on most surfaces (prosthetics catheters, medical equipment, hospital ventilation ducts, open wounds, food, etc.) [5,6], they have become a potential risk. Thus, bacteria MR infections are considered one of the three primary health problems of the 21st century, which will cause 300 million deaths with an economic loss greater than USD 100 trillion in the next 28 years [7].

Nanotechnology has contributed to finding solutions to this problem, developing antibacterial nanomaterials, e.g., carbon-based nanomaterials such as graphene oxide, reduced graphene oxide, carbon dots, and carbon nanotubes [5,8]. Additionally, metal nanoparticles (NPs) such as AgNPs, AuNPs, PtNPs, etc. [9], and metal-oxide-based nanomaterials including magnetite (Fe_3_O_4_) NPs, maghemite (γ-Fe_2_O_3_) NPs [10], nanocomposites (NCs) of Ag-TiO_2_ or hydroxyapatite (HAp) doped with magnetic ions (Sr/Fe) have been employed to combat this issue due to their intrinsic antibacterial activity, and, according to their nature, they can inhibit Gram-positive and Gram-negative bacteria by different mechanisms [1,11,12,13]. For example, Fe_3_O_4_-NPs, NCs of Ag-TiO_2_, have shown antibacterial activity against *Staphylococcus aureus* (*S. aureus*) and *Escherichia coli* (*E. coli*) depending on their concentration and high reactive oxygen species (ROS) generation capacity [14,15]. Likewise, NCs of HAp and Fe_3_O_4_ work against *Micrococcus luteus* and *E. coli* [16]. In several cases, the antimicrobial activity of the cited nanomaterials can be enhanced with functionalizing agents (antibiotics, polymers, organic molecules, etc.) [17,18,19] or by incorporating therapies such as photodynamic therapy (PDT) and PTT [20,21,22].

To obtain these nanomaterials, green synthesis (GS) has been proposed, which presents notable advantages because when using plant extracts from leaves, fruits, seaweeds, and so on [23], organic molecules are available that take the role of reducing, capping, and stabilizing agents [24,25] at once, and depending on the green extract, it can provide molecules with antibacterial properties. For example, Potbhare et al. [26] synthesized CuO nanospheres using *Phyllanthus reticulatus*, where their nanospheres enhanced bacterial inhibition in *E. coli*, *Klebsiella pneumoniae*, and *S. aureus* strains. In addition, the GS of MNPs-Fe proposes reducing or replacing the conventional agents that are dangerous in their handling and have toxic composition, e.g., ammonium hydroxide (NH_4_OH) or sodium hydroxide (NaOH) [27,28,29,30,31], typically used in MNPS-Fe synthesis, such as coprecipitation [32], sol-gel [33], hydrothermal [34], or solvothermal methods [35], among others [36,37,38]. Hence, the GS becomes easy and environmentally friendly [39].

The use of plant extracts in the GS of MNPs-Fe incorporates the circular economy concept due to the use of organic waste from the food industry [40]. Likewise, GS allows incorporating heating by microwave (MW) irradiation, achieving higher reaction rates and thus reducing synthesis times due to rapid and homogeneous volumetric heating (rotation of electric dipoles and ions) compared to conventional heating methods, e.g., heating by flame or electric resistance [41,42,43,44].

On the other hand, antibacterial PTT uses light irradiation, typically in the near-infrared (NIR, 700–1350 nm, optical window) [45], to raise the temperature to ~50 °C of PTAs [46,47], which host bacteria cell membranes or internalize by several mechanisms such as passive diffusion or receptor-mediated endocytosis, among others [1,48,49], causing irreversible damage in the cell membrane and protein disruption, which leads to bacterial death [20]. When using MNPs-Fe as PTAs, a characteristic effect is achieved, known as localized surface plasmonic resonance (LSPR). The electric field of NIR light induces collective resonant oscillation of free electrons (electron cloud) of the MNPs-Fe, due to the absorbed photons, which are converted into phonons, increasing the temperature in the crystal lattice of the MNPs-Fe [47]. For this reason, MNPs-Fe are excellent candidates as broad-spectrum antibacterial agents in antibacterial PTT [4,50].

Regarding cytotoxicity, the toxic effect of MNPs-Fe at a cellular level has been studied in healthy eukaryotic cells (e.g., fibroblasts, liver cells, stem cells derived from adipose tissue, etc.) [51,52] and diseased cells (mainly cancer cells, e.g., DU -145, LNcaP, HeLa, MCF-7, etc.) [10,53,54], as well as in prokaryotic cells, e.g., *S. aureus*, *E. coli*, and so on (a toxic effect on bacteria known as antibacterial activity) [13,18,55]. In any case, the cytotoxic effect depends on the concentration of the MNPs-Fe. Similar mechanisms of cell damage are present in prokaryotic and eukaryotic cells. The MNPs-Fe induce oxidative stress by generating reactive oxygen species (ROS). It releases Fe ions (Fe^+2^, Fe^+3^) that create an acidic environment and, in the presence of H_2_O_2_, produce ^•^OH radicals (Fenton reaction) that depolymerize polysaccharides and inactivate enzymes, lipid peroxidation, and DNA damage. MNPs-Fe can also cause mechanical disruption by localizing to the cell membrane [18,56]. However, in the human body, endocytosis is a crucial mechanism by which cells interact with large or small particles. The phagocytes use their plasma membrane to engulf NPs (phagocytosis) or, in turn, are inhibited by forming a crown of non-specific proteins, producing unpredictable pharmacokinetics [57]. Thus, their size, morphology, and surface chemistry play an essential role in the interaction with target cells; e.g., the kidneys remove NPs of a size < 10 nm and the liver removes NPs of a size > 10 nm, which can also be phagocytized [33,34].

Biotherapies, such as oncology, seek to improve biodistribution times to reach the target cells. Thus, to improve the availability of NPs in the biological environment, they typically are functionalized with polymeric molecules such as polyethylene glycol (PEG), which produces a protein repulsion effect, or molecules of interest that allow increasing selectivity toward target cells [58]. In this sense, GS is also beneficial due to the great availability of organic molecules. In the case of MNPs-Fe, their superparamagnetism (SPM) behavior makes the magnetic targeting towards the required zone possible. Certainly, the surface chemistry and size of the NPs govern their interaction with the target cell [18]. In the case of MNPs-Fe with SPM behavior, it makes the targeting toward the required zone possible. To achieve SPM behavior, uniform size distribution (<100 nm) of the MNPs-Fe and a non-interacting system are essential, i.e., an adequate concentration of MNPs-Fe that does not allow the formation of agglomerates and the MNPs-Fe do not interact with each other in close proximity [59]. Avoiding the agglomeration of NPs is an intrinsic challenge in the synthesis routes of nanomaterials and makes it difficult to adequately develop bio-applications [60].

The current study proposes a fast and simple GS to obtain MNPs-Fe. MW irradiation, iron precursors, ammonium hydroxide, and orange peel extract were employed to achieve this aim. Regarding the organic extract, it came from waste produced by street orange juice vendors. The organic compounds within the orange peel extract constitute reducing agents in combination with low amounts of ammonium hydroxide. To compare the properties of the as-prepared MNPs-Fe, we also carried out different syntheses using the coprecipitation (CO) method and irradiating with MW without the organic extract. We evaluated the produced weight of GS, the antibacterial activity within PTT, cytotoxicity, SPM behavior of the MNPs-Fe, and the effects of the incorporation of organic molecules into de MNPs-Fe compared to others obtained by CO and MW.

## 2. Results and Discussion

### 2.1. Weight Obtained of MNPs-Fe

Quantification of the yield in mass is essential to evaluate the potential scaling of the synthesis at an industrial level. To obtain the mass yield of the GS, CO, and MW syntheses (from the weight of the precursors and the MNPs-Fe obtained, see Table 1), the MNPs-Fe samples were dried (see Section 2.3), and triplicate tests were carried out. Figure 1 presents the average weights obtained for each synthesis. This information is detailed in Appendix A. These results indicated that GS produced a higher weight of MNPs-Fe for NH_4_OH (% v/v) between ~50 and 100% v/v. The 100GS-MNPs-Fe and 100MW-MNPs-Fe samples are the same, i.e., they had 100% v/v of NH_4_OH and were synthesized under the same conditions without orange peel extract. In the concentration range of NH_4_OH (% v/v) 0 to ~50% v/v, the GS produced weight is considerably reduced compared to CO and MW, which maintain their produced weight. However, the role of the organic molecules of the orange peel extract as a reducing agent stands out because, in the absence of NH_4_OH, 0GS-MNPs-Fe were achieved in small quantities (8.97 ± 2.32 mg). In turn, at 0% v/v of NH_4_OH in CO, MW did not crystallize MNPs-Fe. Only DI water was added to the precursor solution, reducing its concentration without generating any reaction.

It is essential to indicate that the magnetic character of the sample 0GS-MNPs-Fe was not stable, i.e., this solution lost its magnetism in short periods (24–48 h). The low stability of 0GS-MNPs-Fe can be justified due to the oxidation of the magnetite and maghemite on the surface of a large number of organic molecules available (100% v/v of orange peel extract), as suggested by Lopez G. et al. [61].

In GS, it is complex to establish a formation reaction of the MNPs-Fe, owing to the great variety of organic compounds. We must consider that the orange peel extract differs according to the place and period of the orange harvest [31,62]. It is known that Fe^+2^ and Fe^+3^ ions form intermediate compounds such as Fe(OH)_2_ and Fe(OH)_3_ to generate Fe_3_O_4_ and water [63]. Hence, the availability of OH^−^ ions in the reducing agent’s solutions is crucial in synthesizing MNPs-Fe. For this reason, the pH (function of the H^+^ concentration) in the reducing agent’s solutions and the byproducts of the synthesis (GS, CO, and MW) were studied (see Appendix A). The acid character of the orange peel extract (pH = 5.15) did not significantly influence the pH of the reducing agent’s solutions in GS, and it remained between 11.59 and 13.16 for solutions between 10 and 100% v/v of NH_4_OH, i.e., a similar availability of hydroxyl ions (OH^−^) in any case (CO and MW). Likewise, GS and MW product pH measurements were more environmentally friendly (pH = 7.20–8.76) than CO (pH = 9.74–11.11). The pH measurements were consistent with the produced weight of GS and MW since the synthesis consumed a more significant amount of OH^−^ ions to form MNPs-Fe. However, in GS, for 0 to ~50% v/v of NH_4_OH, a significant amount of available organic molecules can interact with the OH^−^ reducing the obtained weight of MNPs-Fe. In the specific case of 0GS-MNPs-Fe, the byproducts were acidic (pH = 1.52) because the few available OH^−^ were consumed, increasing the concentration of H^+^.

Equation (1) proposes a possible mechanism for the formation of MNPs-Fe by GS according to the precursors and reducing agents used in this work, where the quantity and type of molecules from the orange peel extract is a factor that influences the weight obtained of MNPs-Fe. Equation (1) was adjusted from the proposal made by Palash D. et al. [39], who used *Lathyrus sativus* peel extract.
(1)$${\mathrm{Fe}}^{+3}+3{\mathrm{Cl}}^{-1}+{\;\mathrm{Fe}}^{+2}+{\mathrm{SO}}_{4}{}^{-2}+{\;\mathrm{OH}}^{-}+{\;\mathrm{NH}}_{4}^{+}+{\mathrm{NH}}_{3}+\mathrm{orange}\;\mathrm{peel}\;\mathrm{extract}\;+H_{2}O\;\overset{\mathrm{Microwave}}{\to }{\mathrm{Fe}}_{3}O_{4}+\gamma -{\mathrm{Fe}}_{2}O_{3}+\mathrm{byproducts}$$

The sample 50GS-MNPs-Fe was selected to develop the cytotoxicity and antibacterial PTT studies for their high weight obtained (88.13 ± 13.33 mg) and low NH_4_OH consumption (greater green approach) compared to samples 75GS-MNPs-Fe (95.03 ± 9.85 mg) and 100GS-MNPs-Fe (84.20 ± 3.10 mg). In addition, its weight obtained was comparable to that of 50MW-MNPs-Fe (80.10 ± 4.61 mg) and 50CO-MNPs-Fe (72.33 ± 8.96 mg), which did not occur for the lower 50% v/v of NH_4_OH samples in GS.

To have comparative results, the mass yield of Fe was calculated from the amount of Fe in the precursors and the Fe present in the obtained MNPs-Fe, and these are shown in Table 1. The results acquired in XRD and EDS were considered.

The mass yields of the synthesis by CO and MW were similar to those reported in analogous studies (62–97%) [64,65,66] that use Fe salt-type precursors. GS generated low yields by mass (1.12–29.36%), except for the 100GS-MNPs-Fe sample (without extract). This stands out with the results reported by E. Nnadozie et al. (3–85%) [67]. However, in the synthesis that we propose in this study, the synthesis time is considerably reduced (a few minutes) compared to the 14 days reported by E. N et al., which includes drying *Chromolaena odorata* leaves, used as a green extract. This is due to using microwaves and vegetable extract in the liquid state. Likewise, reducing the amount of magnetic phase allows the incorporation of organic molecules, which provide additional benefits to the as-synthesized MNPs-Fe.

### 2.2. MNPs-Fe Characterization

#### 2.2.1. Morphology, Size Distribution, and Topography of MNPs-Fe

The SEM images of the samples obtained by GS allowed us to identify a quasi-spherical morphology and a normal size distribution; see Appendix A. The 50GS-MNPs-Fe sample had an average size of 49.3 ± 9.6 nm, the smallest of the GS samples (see Figure 2a). A few agglomerates were also found and neglected to determine the average size of the MNPs-Fe. DLS analysis allowed the study of the hydrodynamic diameter of the 50GS-MNPs-Fe sample. Thus, two populations of different sizes were observed; see Figure 2b. In DLS, a model based on the intensity of the scattered electric field for polydisperse particles was used, which presents intensity contributions according to the hydrodynamic radius of the populations present. Therefore, the highest intensity does not refer to the population quantity of MNPs-Fe [68]. This result corroborates that the few agglomerates observed in SEM comprised the 224.8 ± 0.4 nm population. The 74.7 ± 0.1 nm population corresponds to the 50GS-MNPs-Fe size coated with organic molecules from the orange peel extract, which is consistent with the study developed by Bano S. et al. [31]. This shell of organic molecules has a thickness of 25.4 nm, as indicated in Figure 2b. It is necessary to highlight that the SEM images show the naked MNPs-Fe.

The AFM topography of the GS samples enabled the study of the average roughness (Ra) of the MNPs-Fe and corroborated a quasi-spherical morphology. Figure 2c corresponds to the topography of the 50GS-MNPs-Fe sample. The contours of the MNPs-Fe (red curve) in two-dimensional profiles along an arbitrary axis (e.g., AB axis in Figure 2c) acquired from Gwyddion 2.60 software empowered us to study Ra. Consequently, 20 random measurements of Ra were achieved in different contours and profiles of the MNPs-Fe, omitting regions of agglomerates and without MNPs-Fe. To estimate a representative Ra of the scanned section, Ra was measured in 20 profiles of the MNPs-Fe. This was achieved by taking AB axes randomly, which comprise different MNPs-Fe, and avoiding agglomerates. The values of Ra increased according to the amount of orange peel extract used in GS; see Appendix A. The 50GS-MNPs-Fe sample showed an Ra = 78 ± 22 pm, while 100GS-MNPs-Fe (without orange peel extract) exhibited an Ra = 58 ± 12 pm. Thus, in GS, the shell of MNPs-Fe formed by organic molecules from the orange peel extract increases the value of Ra. Furthermore, it is essential to consider that the Ra values are sensitive to pixel resolution and size [69]. Likewise, Ra can be affected by the interaction between the AFM tip (Tapping mode) and the organic molecules bound to the MNPs-Fe since this superficial organic crown is approximately 24.2 nm (DLS) and can be considered a soft sample, as suggested by Christensen M. et al. [70].

#### 2.2.2. Phase and Chemical Composition of the MNPs-Fe

The diffraction patterns of the MNPs-Fe, obtained by GS, MW, and CO, presented phases of magnetite (Fe_3_O_4_) and maghemite (Fe_2_O_3_). Their relative peak positions (2θ) most coincided with crystallographic charts reported in the Crystallography Open Database (COD) to magnetite; see Appendix A–c. Nevertheless, the characteristic peaks of the organic molecule shell in these samples could reduce the intensity of the characteristic peaks of magnetite [17]. The XRD patterns of the samples obtained by CO and GS presented relative peak positions and intensities coinciding with the COD crystallographic charts because these MNPs-Fe do not have an organic shell; see Appendix A.

Figure 3a shows the XRD patterns of 50GS-MNPs-Fe, 50MW-MNPs-Fe, and 50CO-MNPs-Fe samples. In the three cases, the characteristic peaks of magnetite and maghemite were observed at 2θ: 18.4° (111), 30.2° (220), 35.5° (311), 37° (222), 43.1° (400), 53.4° (242), 57.1° (511), 62.8° (440), 74.1° (533) and 89.5° (713) [71,72], matching with the crystal charts of COD. The 50GS-MNPs-Fe sample corresponded to magnetite COD ID 1011032 (cubic system, Fd-3m) and maghemite COD ID 9006316 (cubic system, Fd-3m) with a contribution of 47.0% and 53.0%, respectively. The 50MW-MNPs-Fe sample was identified as magnetite COD ID 9005842 (cubic system, Fd-3m) and maghemite COD ID 1528612 (cubic system, P4_1_32), with a contribution of 91.2% and 8.8%, respectively. Likewise, the 50CO-MNPs-Fe sample coincided with magnetite COD ID 9002319 (cubic system, Fd-3m) and maghemite COD ID 9006317 (cubic system, P4_1_32) in a contribution of 63.5% and 36.5%, respectively. This identification was carried out in the Match! 3 software. See Table 1 for all samples. The contribution of magnetite, maghemite, and hematite in the 0GS-MNPs-Fe, 25GS-MNPs-Fe, 50GS-MNPs-Fe, and 75GS-MNPs-Fe samples are not absolute since they have a contribution of organic molecules from the green synthesis.

The XRD pattern of 50GS-MNPs-Fe shows many additional peaks. These peaks might correspond to organic molecules of the orange peel extract (MNPs-Fe shell), which have a crystalline structure, for example, crystalline cellulosic components with characteristic peaks at 2θ: 15°, 22° and 35° [73]. Likewise, compounds such as Linalool and Eugenol reported in orange peel [31,62] have characteristic peaks at 2θ: 11.7°, 15.7°, 17.2°, 21.4°, 22.5°, 23.6° and 30.2° [74]. The crystallite size (D_c_) was calculated using the Scherrer equation [75], and the FWHM (full width at half maximum) value was obtained using the OriginPro2018 software using a Voigt-type nonlinear fit on the peaks corresponding to the planes (131) (greater intensity). The Scherrer constant k = 0.89, typical for spherical geometries, was considered. The D_c_ of the 50GS-MNPs-Fe, 50MW-MNPs-Fe, and 50CO-MNPs-Fe samples was 10.97 nm, 10.25 nm, and 7.84 nm, respectively. Appendix A presents D_c_, lattice parameters, FWHW, and 2θ corresponding to all the samples described in Table 1. D_c_ refers to coherent diffracting crystalline domains as indicated by various studies [76,77,78]. Therefore, the smaller D_c_ size than the sizes obtained by SEM suggests that the MNPs-Fe are composed of several crystallites generated by stacking defects in the nucleation of the MNPs-Fe [76].

The EDS elemental analysis (Figure 3b) allowed identifying the presence of carbon (C), oxygen (O), and iron (Fe) in the 50GS-MNPs-Fe sample. The high weight percentages of C (48.847 ± 6.087%), O (29.298 ± 3.7%), and Fe (21.487 ± 0.666%) were consistent with the XRD results, i.e., magnetite, maghemite phases and a shell of carbon-based organic molecules. In the samples obtained by GS, the C content (wt. %) increased according to the amount of orange peel extract used in the synthesis; see Appendix A.

Raman spectra were acquired in the range of 100–2000 cm^−1^. Nonetheless, the active Raman bands for magnetite and maghemite are within 100–800 cm^−1^ [46,47]. Therefore, Figure 3c shows the Raman spectrum of the 50GS-MNPs-Fe sample in 100–800 cm^−1^. The gray regions correspond to characteristic Raman bands of magnetite, and the light blue ones to characteristic bands of maghemite [79,80,81]. Active bands of magnetite were observed at 197 cm^−1^ ($T_{2g}^{1}$, translational movement of the Fe_3_O_4_ structure), 297.2 cm^−1^ ($T_{2g}^{2}$, asymmetric stretching of Fe and O), 560.5 cm^−1^ ($T_{2g}^{3}$, asymmetric bending of O with respect to Fe), and 668.1 cm^−1^ ($A_{1g}$, symmetric stretching of O along Fe-O bonds) consistent with reports of active bands of magnetite [36]. Likewise, the maghemite active bands observed were 343 cm^−1^ ($T_{1}$, asymmetric stretching Fe-O), 490.4 cm^−1^ (E, symmetric bending O along Fe-O bonds), and 698 cm^−1^ ($A_{1}$, symmetric stretching Fe-O) similar to the bands reported in [82,83]. The bands 125–135 cm^−1^ correspond to the vibration of crystalline structures (LA-Modes) [84]. The band observed at 413.7 cm^−1^ was attributed to the symmetrical stretching between O external to magnetite and maghemite with Fe ions belonging to iron oxides [84]. The Raman spectrum of the 50GS-MNPs-Fe sample was compared to magnetite (record ID 319 obtained from the HORIBA database), generating a 79.32% correlation with the help of Wiley Science’s KnowItAll software.

The complete Raman spectrum (see Appendix A) allowed identifying active bands of molecule bonds that cover the 50GS-MNPs-Fe sample (derived from orange peel extract). Appendix A shows the Raman active bands observed, with their molecular vibrations according to atomic bonds and the possible molecules present.

Figure 3d shows the FT-IR spectrum of the 50GS-MNPs-Fe sample. Active bands were observed between 600–450 cm^−1^, corresponding to vibrations between Fe-O atoms of the magnetite and maghemite phases [85,86,87,88]. Organic molecules’ active FT-IR bands were also observed and are shown in Appendix A. The FT-IR active band at 2532 cm^−1^ (S compounds) can be generated by ions (SO_4_^2−^) derived from Fe^2+^ sulfate. Likewise, the halogens are due to ions (Cl^−^) from Fe^3+^ chloride used as a precursor. Additionally, the 2,350 cm^−1^ band of amine salts suggests the formation of compounds derived from NH_4_OH.

Appendix A were developed according to the active bands reported by Horiba J. [84] for the analysis of Raman and FT-IR spectra, as well as the studies carried out by S. Bano et al. [31] and G. Kahrilas et al. [62].

Using FT-IR and Raman spectroscopy, the presence of magnetite and maghemite was corroborated, as well as the functionalization of the 50GS-MNPs-Fe with organic molecules of the type alcohols, aldehydes, S compounds, amines, ketones, aromatic rings, halogens, and terpenes.

#### 2.2.3. Magnetometry

Figure 4a shows M vs. H curves of the samples 50GS-MNPs-Fe, 50MW-MNPs-Fe, and 50CO-MNPs-Fe at different temperatures (60 K, 62 K, 65 K, and 300 K). The zoom at 300 K (upper left box) indicates almost zero remanent magnetization (Mr), as well as a coercivity (Hc) close to zero in all samples. The zoom at 60 K (lower right box) indicates higher Mr and Hc in the 50MW-MNPs-Fe and 50CO-MNPs-Fe samples. Nonetheless, the Mr and Hc of the 50GS-MNPs-Fe sample remain close to zero. This behavior and the Langevin-type curves suggest that at 300 K, all the samples exhibit an SPM behavior, and by reducing the temperature to 62 K and 65 K in the 50CO-MNPs-Fe and 50MW-MNPs-Fe samples, respectively, cross the blocking temperature (T_B_) to a blocked state. Even so, the 50GS-MNPs-Fe sample maintains its SPM behavior, where the magnetic phase corresponded to 4.05 g of magnetite and maghemite in the sample and the rest (9.35 g) to the coating of organic molecules. Figure 4b indicates that reducing the temperature in the 50CO-MNPs-Fe and 50MW-MNPs-Fe samples increases Hc, indicating the opening of the M vs. H magnetization curves, which generates hysteresis. In contrast, in the 50GS-MNPs-Fe sample, the M vs. H curve remains close with SPM behavior.

The Ms of the 50MW-MNPs-Fe sample is the highest, being 65 emu·g^−1^ at 300 K and 75.4 emu·g^−1^ at 65.5 K, followed by the values of the 50CO-MNPs-Fe sample (50.5 emu·g^−1^ at 300 K, 60.7 emu·g^−1^ at 62 K), and finally those from the 50GS-MNPs-Fe sample with a Ms of 44.1 emu·g^−1^ at 300 K and 72.8 emu·g^−1^ at 60 K. This result can be explained due to the 50MW-MNPs-Fe sample having the greatest magnetite phase (91.2%) as observed in XRD, followed by the 50CO-MNPs-Fe sample (63.5%) and then by the 50GS-MNPs-Fe sample (47.0%), which is consistent with the reported Ms of magnetite in the bulk state (90–92 emu·g^−1^ at 300 K) [89,90,91] and maghemite (30–80 emu·g^−1^ at 300 K) [92,93]. This reduction in Ms is also affected by the greater surface anisotropy (nanometric scale). Even so, it should be noted that the Ms of the 50GS-MNPs-Fe presented a significant increase at 60 K because the SPM behavior was maintained (see Figure 4a). It is also important to note that MNPs-Fe with several coherent diffracting crystalline domains (suggested by D_c_ in XRD) reached monodomain magnetic (SPM) status. This can occur due to the energetic ease of aligning the atomic magnetic moments in the synthesized MNPs-Fe of sizes around 49.3 nm, generating a maximum net magnetic moment for each nanoparticle.

Figure 4c presents the ZFC and FC curves of the 50GS-MNPs-Fe sample and indicates that the temperature of T_B_ (transition from an SPM state to a blocked state) is below 60 K since no inflection is observed in the ZFC curve (operating capacity of the VSM), taking the criterion that the maximum peak of the ZFC curve corresponds to T_B_ [94]. The irreversibility temperature (T_ir_) identified at 275 K (superposition of ZFC and FC curves) suggests a broad size distribution of the 50GS-MNPs-Fe [95], which is consistent with the presence of aggregates in the samples. The T_B_ of the 50MW-MNPs-Fe and 50CO-MNPs-Fe samples were determined at 211.1 K and 160.09 K, respectively. See Appendix A for more information. Therefore, the 50GS-MNPs-Fe with SPM behavior can be used in applications that occur at temperatures greater than 60 K and below the Curie temperature; it must also be added that the terpenes, aldehydes, and other organic molecules (coating around 25 nm) prevent the aggregation of the 50GS-MNPs-Fe with a size of ~49.3 nm, quasi-spherical, with a magnetic nucleus (magnetite and maghemite), which improves its performance in bioapplications.

### 2.3. Cell Viability in ATCC RAW 264.7 Cells

Figure 5a,b show the ATCC RAW 264.7 cell viability percentage under different concentrations (3.9–2000 µg·mL^−1^) of MNPs-Fe (50GS-MNPs, 50MW-MNPs-Fe, and 50CO-MNPs-Fe) for 24 h and 8 days, respectively.

The percentage of cell viability of ATCC RAW 264.7 cells under concentrations up to 250 µg·mL^−1^ of MNPs-Fe (GS, MW, and CO) did not present significant changes at 24 h. The prolonged cell viability assays (8 days) allowed us to corroborate the biocompatibility of the MNPs-Fe [96,97,98], up to concentrations between 62.5 and 250 µg·mL^−1^ of MNPs-Fe. The IC50 (inhibition of 50% of the cells) obtained at 8 days with the samples 50GSMNPs-Fe, 50MW-MNPs-Fe, and 50CO-MNPs-Fe was 461.4 µg·mL^−1^, 217.7 µg·mL^−1^, and 520.2 µg·mL^−1^, respectively, while at 24 h, IC50 values of 318 µg·mL^−1^, 1274 µg·mL^−1^, and 601.6 µg·mL^−1^ were obtained with the 50GSMNPs-Fe, 50MW-MNPs-Fe, and 50CO-MNPs-Fe samples, respectively. IC50 was calculated in GraphPad Prism software with a 95% confidence interval and a four-parameter logistic fit (nonlinear least squares regression).

The cell viability of ATCC RAW 264.7 cells under concentrations up to 250 µg·mL^−1^ of MNPs-Fe (GS, MW, and CO) did not present significant changes (80–100%) at 24 h. In contrast, after 8 days, the 50MW-MNPs-Fe and 50CO-MNPs-Fe samples did not show significant differences in cell viability (80–100%) for concentrations lower than 62.8 µg·mL^−1^, while the 50GS-MNPs-Fe was maintained at 250 µg·mL^−1^ (80–100% cell viability). This result is consistent with other studies with low concentrations (100–300 µg·mL^−1^) of MNPs-Fe [99,100], where negligible cytotoxicity is reported.

These data can be confusing. However, these results can be elucidated by considering the state of the aqueous solutions studied by DLS (see Appendix A). The 50MW-MNPs-Fe sample presents more aggregates (~228.4 nm, ~1033 nm), followed by the 50CO-MNPs-Fe sample (~166.2 nm, ~726.2 nm). Therefore, the number of MNPs-Fe dispersed and available for internalization in the ATCC RAW 264.7 cells is reduced, which leads to less cytotoxicity. Thus, in 24 h, the 50GS-MNPs-Fe sample is apparently more cytotoxic. Still, in reality, we could say that they are better dispersed with fewer aggregates in such a way that there are more MNPs-Fe internalized in the macrophage cells, causing oxidative stress. Then, as the test days go by (day 2, day 3, etc., until day 8), the naked 50MW-MNPs-Fe and 50CO-MNPs-Fe samples (only with magnetite and maghemite phases identified by XRD, without organic coating) degrade as a source of Fe^+2^ and Fe^+3^ ions towards the cell culture medium. These ions are highly reactive and cause oxidative stress and, therefore, more significant cytotoxicity. This result explains the dramatic change in IC50 at 24 h and 8 days in 50MW-MNPs-Fe. In samples with 50GS-MNPs-Fe, those mentioned above can also occur, but to a lesser extent; that is, the organic coating prevents their aggregation (aggregates up to~224.8 nm), as observed in DLS Appendix A. This agrees with the work of Guldrris et al. [101], who used MNPs-Fe functionalized with poly(acrylic acid) and MSCs (mesenchymal stem cells), which maintain viability for 12 days.

These results suggested that in short periods (24 h), 50MW-MNPs-Fe and 50CO-MNPs-Fe have lower cytotoxicity. That is, 50% of ATCC RAW 264.7 cells maintained their normal mitochondrial metabolic function at around 1274 and 601.6 µg·mL^−1^ concentrations for 50MW-MNPs-Fe and 50CO-MNPs-Fe, respectively. In contrast, the 50GS-MNPs-Fe sample achieved an IC50 of 318.2 µg·mL^−1^. That is, a lower concentration of MNPs-Fe caused more significant cell death. In more prolonged periods (8 days), the IC50 induced by the 50GS-MNPs-Fe and 50CO-MNPs-Fe samples were higher (461.4 and 520.2 µg·mL^−1^), while the 50MW-MNPs-Fe samples were more cytotoxic. (217.7 µg·mL^−1^).

Figure 5c presents images taken with a confocal microscope of the Giemsa staining assay, with ATCC RAW 264.7 cells at concentrations of 450 µg·mL^−1^, 200 µg·mL^−1^, and 500 µg·mL^−1^ of 50GS-MNPs-Fe, 50MW-MNPs-Fe, and 50CO-MNPs-Fe samples, respectively. The concentrations in the Giemsa staining assay were approximated to the IC50 calculated at 8 days in the Resazurin cell viability assay. The Giemsa staining assay enabled the study of the damage and changes produced by MNPs-Fe in ATCC RAW 264.7 cells. At 24 h, the 50GS-MNPs-Fe sample reduced the size of the ATCC RAW 264.7 cells compared to the control. At 8 days, a cell size like that of the control was observed, as well as a small leak of the cytoplasm (red arrow). At 24 h and 8 days, chromatin condensation was observed without significant changes in the morphology of the cell membrane, which indicates death by apoptosis. A lack of nutrients or excess ROS can cause this programmed cell death. ROS can induce mitoptosis death, which arises from removing ROS-overproducing mitochondria by autophagy [102].

In contrast, the samples with 50MW-MNPs-Fe induced a reduction in cell size and significant cytoplasm leakage after 24 h and 8 days. Likewise, the 50CO-MNPs-Fe samples caused severe damage to the cell membrane and the leakage of the cytoplasmic elements (red arrows). Both types of MNPs-Fe (CO and MW) produced a necrosis-like death, a more aggressive and uncontrolled death.

Thus, 50GS-MNPs-Fe presents advantages over those obtained by CO and MW in terms of biocompatibility since the organic molecules of the orange peel prevent their aggregation and induce cell death by apoptosis in higher concentrations.

### 2.4. Photothermal Studies

Figure 6 depicts the photothermal study of aqueous solutions with 50GS-MNPs-Fe, 50MW-MNPs-Fe, and 50CO-MNPs-Fe at concentrations of 8 mg·mL^−1^, including the control sample of DI water. All samples (in triplicate) were irradiated under the red light of 630 nm and 65.5 mW·cm^−2^ from an LED source. On the horizontal axis of Figure 6, the doses of light supplied are linked according to the irradiation times (increments of two minutes) with a total of 44 min of irradiation. All the MNPs-Fe solutions indicated that the temperature reaches a temperature threshold of 30 min. The 50GS-MNPs-Fe, 50MW-MNPs-Fe, and 50CO-MNPs-Fe reached maximum temperatures of 50.8 ± 0.2 °C, 51.7 ± 0.6 °C, and 49.6 ± 0.71 °C, respectively, while the control sample (DI water) reached 42.3 ± 0.1 °C. The photothermal effect was due to this type of nanoparticle LSPR [53]. These results indicate that all samples are suitable PTAs in the PTT for eliminating bacterial and/or tumor cells [17,55,103,104,105,106,107].

It is essential to highlight that the 50GS-MNPs-Fe sample achieved a photothermal effect similar to MW and CO, despite having a lower amount of magnetite and maghemite phase. The concentrations for this study were obtained with the weight obtained after the synthesis.

### 2.5. Antibacterial PTT

Figure 7a,b show the Log_10_ (CFU) count of the spread plate assays in *S. aureus* and *E. coli* bacteria, respectively. All groups were treated with MNPs-Fe (GS, MW, and CO) using a concentration of 8 mg·mL^−1^ of MNPs-Fe in PBS. The antibacterial activity of 50GS-MNPs-Fe against *S. aureus* without irradiation is small in 0.25 Log_10_ (CFU). Likewise, 50MW-MNPs-Fe and 50CO-MNPs-Fe without irradiation did not have significant antibacterial activity. Irradiation with a red light at 630 nm (65.5 mW·cm^−2^) for 30 min towards *S. aureus* (without MNPs-Fe) had no antibacterial effect. However, 50GS-MNPs-Fe and 50CO-MNPs-Fe with red light irradiation at 630 nm (65.5 mW·cm^−2^) for 30 min completely inhibited *S. aureus* bacteria. The 50MW-MNPs-Fe with red light irradiation at 630 nm (65.5 mW·cm^−2^, 30 min) inhibited 3.99 Log (CFU) of *S. aureus*.

The antibacterial activity of all MNPs-Fe (GS, MW, and CO) against *E. coli* is negligible. The 50MW-MNPs-Fe irradiated with a red light at 630 nm (65.5 mW·cm^−2^, 30 min) completely inhibited *E. coli*, while the 50GS-MNPs-Fe and 50CO-MNPs-Fe were able to inhibit 2.54 Log_10_ (CFU) and 4.19 Log_10_ (CFU), respectively (see Figure 8b for 50GS-MNPs-Fe against *E. coli*). The results indicated that 50GS-MNPs-Fe and 50CO-MNPs-Fe have excellent antibacterial activity against *S. aureus* (Gram positive) (see Figure 7a and Figure 8a), while 50MW-MNPs-Fe are capable of inhibiting Gram-negative bacteria as *E. coli*; see Figure 7b and Figure 8c.

The photothermal effect of the MNPs-Fe contributed to bacterial elimination. However, it is highlighted that in laser irradiation, MNPs-Fe act as a photosensitizing agent (PS) that increases the generation of ROS (singlet oxygen, etc.), which causes oxidative stress and cell death. Therefore, bacteria inhibition is due to the contribution of the photothermal effect along with ROS generation that produces a photodynamic effect, as proposed in several studies [21,108,109,110].

Figure 8b shows that the final temperature after the irradiation (antibacterial PTT) was higher than 50 °C for both bacterial strains, while the controls of light (bacteria + light) reached temperatures around 40 °C. In all the irradiated cases, the dose of light supplied corresponds to 117.9 J·cm^−2^.

The results of the antibacterial activity indicate that the 50GS-MNPs-Fe inhibit *S. aureus* and partially *E. coli*, which is in accordance with [71]. However, the concentrations of MNPs-Fe used in our studies are higher, reaching temperatures of ~50 °C in antibacterial PTT, similar to those reported by Qing G. et al. [111] from 43 to 49 °C (with 808 nm, 0.5 W·cm^−2^).

## 3. Materials and Methods

### 3.1. Materials

Iron III chloride hexahydrate (FeCl_3_·6H_2_O) was from Mallinckrodt Chemical Ltd. (Chesterfield, UK). Iron II sulfate heptahydrate (FeSO_4_·7H_2_O) was from MERCK (Rahway, NJ, USA). Ammonium hydroxide (NH_4_OH) was from BDH Chemicals (Welch, WV, USA). DMEM-Dulbecco’s Modified Eagle Medium, agar, and broth culture was from Fisher Scientific (Princeton, NJ, USA). Resazurin sodium salt was from Sigma Aldrich (St. Louis, MO, USA). Giemsa Stain was from MERK (RJ, Brazil). No further purification was performed on the chemical agents.

### 3.2. Preparation of Orange Peel Extract

First, the endocarp and inner mesocarp were removed from the orange waste to obtain only the orange peel (outer mesocarp). Then, small pieces approximately 1 × 2 cm in size were obtained. A total of 25 g of orange peel was weighed and thoroughly washed using deionized (DI) water. The orange peel and 250 mL of DI water were blended using a domestic blender at 2000 rpm for 10 min (Oster-4172, 375 W, Acuña, Mexico). The mixture was irradiated in a domestic MW oven (General Electric, JES-700 W, Boston, MA, USA) for 5 min. Finally, the mixture was cooled and sieved using paper filters (Brew Rite, Little Chute, WI, USA). The vegetal extract was used within the next 30 min.

### 3.3. GS of MNPs-Fe

FeCl_3_·6H_2_O (210 mg) and FeSO_4_·7 H_2_O (105 mg) were dissolved in 10 mL DI water in a beaker at 70 °C with continuous magnetic stirring. Then, several mixtures of 25 mL of orange peel extract with NH_4_OH were prepared in different volume percentages: 100%, 75%, 50%, 25%, and 0% NH_4_OH. A solution of FeCl_3_·6H_2_O and FeSO_4_·7H_2_O was added to each mixture of orange peel extract and NH_4_OH and subsequently irradiated with MW (General Electric, JES-70W) for three intervals of 30 s. The black appearance indicated the formation MNPs-Fe, which were washed with DI water and under a magnetic field to remove phytochemicals excess and non-magnetic particles. The MNPs-Fe obtained with 100%, 75%, 50%, 25%, and 0% of NH_4_OH are called 100GS-MNPs-Fe, 75GS-MNPs-Fe, 50GS-MNPs-Fe, 25GS-MNPs-Fe, and 0GS-MNPs-Fe, respectively.

To compare the produced weight of the GS, cytotoxicity, antibacterial activity, and magnetic properties such as saturation magnetization (Ms) and blocking temperature (T_B_), MNPs-Fe were prepared by CO and MW routes without orange peel extract. In the case of CO synthesis, a procedure similar to that described in [112,113,114] was followed, using FeCl_3_·6H_2_O (210 mg) and FeSO_4_·7 H_2_O (105 mg) as precursors with the identical aliquots used in GS. Mixtures according to the percentages used in the GS were prepared with DI water and NH_4_OH. Then, the temperature was raised to 80 °C on a hot plate. Later, the solution of precursor salts was added drop by drop. Magnetic stirring was maintained for 1 h, and then the NPs were washed with DI water and a permanent magnet. The MW synthesis followed the same procedure as the GS [31,115], replacing the orange peel extract with DI water. The samples obtained by CO with 100%, 75%, 50%, and 25% of NH_4_OH (volume) were called 100CO-MNPs-Fe, 75CO-MNPs-Fe, 50CO-MNPs-Fe, and 25CO-MNPs-Fe, respectively. The samples obtained by MW synthesis were denominated as 100MW-MNPs-Fe, 75MW-MNPs-Fe, 50MW-MNPs-Fe, and 25MW-MNPs-Fe, according to the percentage of NH_4_OH used.

In the different tests, samples of MNPs-Fe were in aqueous dispersions and powder form (dry). In the case of MNPs-Fe powder, drying was carried out in an oven at 40 °C for 42 h.

### 3.4. MNPs-Fe Characterization

The morphology and size distribution of the MNPs-Fe were studied by scanning electron microscopy (SEM) in a Tescan Mira 3 microscope (Tescan, Brno, Czech Republic) equipped with a Schottky field emission source at 25 kV (JEOL Ltd., Tokyo, Japan). For this, an aliquot of MNPs-Fe was dispersed in 1 mL of DI water and then dried on a sample holder at room temperature. To measure the hydrodynamic diameter of the MNPs-Fe, a dynamic light scattering (DLS) analysis was performed in a Particle Size Analyzer 90Plus-11554 (Brookhaven Instruments Corporation, Suffolk, NY, USA). Thus, 5 mL of dispersion was used for the sample of MNPs-Fe in DI water at 1 mg·mL^−1^. The topography of MNPs-Fe was studied by atomic force microscopy (AFM) in a Dimension Icon Atomic Force Microscope (Bruker, Billerica, MA, USA), in Tapping Mode with a scanning range of 90 µm × 90 µm (XY) and 10 µm (Z) equipped with a Dim 4000 type scanner. An aliquot of MNPs-Fe on DI water was used at 10 mg·mL^−1^.

The elemental composition was determined by energy dispersive X-ray spectroscopy (EDS) in the same Tescan Mira 3 microscope using a detector Xflash^®^ 6–30 (Bruker, Billerica, MA, USA), with a resolution of 123 eV in Mn Kα. The crystalline structure and phases of iron oxide in the MNPs-Fe were identified by X-ray diffraction (XRD) in an Empyrean diffractometer (Malvern Panalytical, Malvern, UK) equipped with a copper X-ray tube (Empyrean Cu LFF, Almelo, Netherlands) and Cu-Kα radiation at 2θ = 5–90° with 45 kV and 40 mA. For this, 93.3–434 mg of MNPs-Fe powder was used. Fourier transform infrared (FTIR) and Raman spectroscopy allowed the study of the molecular vibrations of MNPs-Fe. FTIR spectra were acquired using a Spectrum 100 (86660) spectrometer (Perkin Elmer, Waltham, MA, USA) equipped with a Spotlight 200 N/S 86672 microscope with an analysis range of 500–4000 cm^−1^ and 5 mg of MNPs-Fe powder. Raman spectra were recorded using a LabRAM HR Evolution spectrometer (HORIBA Corporation, Kyoto, Japan) with an analysis range between 100 and 2000 cm^−1^ and a sensitivity of 10 cm^−1^. For this, 5 mg of MNPs-Fe were dispersed in 5 mL of DI water, and an aliquot was dried on a glass slide at room temperature. 

The magnetic behavior of MNPs-Fe was analyzed in a Quantum Design Versalab Vibrating Sample Magnetometer (VSM) (Quantum Design, San Diego, CA, USA) with an operating range of −3 to 3 Tesla (−30 to 30 kOe) and 50–400 K with sensitivity ~1 × 10^−6^ emu. Thus, this generated two types of curves: magnetization (M) as a function of the external magnetic field (H) at a constant temperature and magnetization (M) as a function of temperature (cooling from 315 K to 60 K) in the presence of zero magnetic fields (zero field cooling, ZFC curves) and with the presence of a small magnetic field of 50 Oe (field cooling, FC curves).

### 3.5. Cytotoxicity Assay

Cytotoxicity assays measure the rate of proliferation and the toxic effects on the cell when using certain materials, such as MNPs-Fe. Particularly, this cytotoxicity study used as a test subject, an animal cell line ATCC RAW 264.7 of the macrophage type from mice (Mus musculus), donated by CISeAL.

#### 3.5.1. Cell Culture

The animal cell line ATCC RAW 264.7 remained in DMEM-Dulbecco’s Modified Eagle Medium accompaniment with 1% penicillin/streptomycin and fetal bovine serum (FBS)–10% (DMEM-10) in a Series II Water Jacket–CO_2_ Incubator (Thermo Scientific, Hillsboro, OR, USA), at 37 °C in an atmosphere of CO_2_ 5% and 98% relative humidity (RH) for 24 h.

#### 3.5.2. Resazurin Cell Viability Assay

Within a 96-well plate with a black background, 2 × 10^4^ cells/well were seeded using 100 µL of DMEM-10 (incubated at 37 °C) with an atmosphere of 5% CO_2_ and 98% of RH for 24 h. MNPs-Fe were diluted in 10-fold serial dilutions (2000, 1000, 500, 250, 125, 62.5, 31.2, 15.6, 7.8, and 3.9 µg·mL^−1^) with DMEM-10. They were then placed in the seeded cells, maintaining the respective life control wells. Thus, 10 µL of resazurin sodium salt at 3 mM in PBS per well was added, and the plate was incubated for 24 h. Finally, fluorescence was measured at 520, 580, and 640 nm excitation on a GloMax (Promega, Madison, WI, USA) multimodal microplate reader. In viable cells with adequate mitochondrial metabolic function, redox reactions occurred between the mitochondrial coenzymes nicotinamide adenine dinucleotide NADH (reduced form) and its oxidized NAD^+^, irreversibly reducing resazurin to resorufin. The resorufin fluoresces at 560 nm, then is excreted to the culture medium, where it was quantified with a plate fluorometer, this signal being a function of the number of viable cells in the sample [116,117]. All samples were carried out in triplicate.

#### 3.5.3. Giemsa Staining Assay

Three sterile circular coverslips per well were aseptically placed into two 6-well plates, and 5.5 × 10^5^ ATCC RAW 264.7 cells per well were plated and incubated at 37 °C with a 5% CO_2_ atmosphere and 98% HR. The following day, cells were washed with PBS, and the MNPs-Fe were added and incubated. A coverslip was removed from each well at 24 h and 8 days, later fixed with 5% of glutaraldehyde for 24 h, and then immersed in Giemsa stain for 30 min. Finally, images were taken with a confocal microscope. This assay was carried out to stain the cell organelles and observe the damage caused by the presence of MNPs-Fe, such as cytoplasmic condensations, damage to the cell membrane, and so on.

### 3.6. Antibacterial Activity

Antibacterial activity studies used *S. aureus* ATCC 25923 and *E. coli* ATCC 25922 bacteria as test subjects (both donated by CISeAL). Light irradiation-assisted PTT was employed as an antibacterial strategy, which converts light energy into heat that kills bacterial cells by raising the local temperature.

#### 3.6.1. Thermal Studies

The photothermal studies measured the temperature increase with a digital thermometer DET-306 (SCANMED, 0.1 °C sensitivity) in MNPs-Fe solutions and DI water at 8 mg·mL^−1^ with red light irradiation of 630 nm from a 65.5 mW·cm^−2^ LED source. The temperature was registered every two minutes until its increase stopped. Three measurement runs were performed for each type of Fe-MNPs and the control.

#### 3.6.2. Bacterial Culture

For bacterial culture, cryovials were provided with *S. aureus* and *E. coli* bacteria that were thawed at room temperature and inoculated in Muller Hinton Broth (Difco^TM^, Princeton, NJ, USA) for incubation (SL Shel lab incubator–1525, San Diego, CA, USA) for 24 h at 37 °C. The number of Colony-Forming Units (CFU) was established using a spectrophotometer (Thermo Scientific ^TM^ Orion^TM^ AquaMate 8000 UV-Vis, Waltham, MA, USA) for absorbance of 0.2 OD and 0.4 OD, equivalent to 10^7^ CFU·mL^−1^ for *E. coli* and 10^6^ CFU·mL^−1^ for *S. aureus,* respectively.

#### 3.6.3. Spread Plate Assay

Four groups were considered for each bacterium: (1) control (bacteria alone), (2) bacterium + light, (3) bacterium + MNPs-Fe, and (4) bacterium + MNPs-Fe + light. All assays were performed in triplicate. Consequently, 1 mL of bacterial culture (*E. coli* and *S. aureus*) was taken and centrifuged at 3,000 rpm for 10 min. Then, the supernatant was discarded, and 1 mL of MNPs-Fe dispersed in PBS was added (8 mg·mL^−1^) or PBS according to the test groups and homogenized with vortex. Each sample was incubated in the dark for 25 min at 37 °C. Then, bacteria + light and bacteria + MNPs-Fe + light groups were irradiated with red light (630 nm) in a LED source of 65.5 mW·cm^−2^ for 30 min. Later, an aliquot of 0.5 mL of each sample was taken to eight serial dilutions (1:2 to 1:256) in PBS to inoculate 4 µL of each dilution into a divided eight-parts Petri dish with Muller Hinton Agar (Difco ^TM^) spread. Finally, the Petri dishes were incubated at 37 °C for 24 h to count the CFU. The temperature reached in the PTT was measured with a digital thermometer DET-306 as soon as the laser irradiation was suspended. Likewise, in a control without irradiation, the temperature was recorded at the end of the incubation time with the MNPs-Fe.

## 4. Conclusions

In this work, MNPs-Fe were synthesized by an environmentally friendly approach using orange residues. These MNPs-Fe were physicochemically and magnetically characterized. The results showed the morphology of nano-spheres coated with organic molecules (from the orange peel). The MNPs-Fe presented crystalline phases of magnetite and maghemite. In addition, an SPM behavior was identified. The GS-MNPs-Fe were compared with others synthesized through CO and MW to study the influence of NH_4_OH on the produced weights of MNPs-Fe. Thus, the minimum aliquot of NH_4_OH was selected (50% v/v) to maintain an adequate produced weight (29.3% mass yield) of GS (sample 50GS-MNPs-Fe). Photothermal studies indicated that 50GS-MNPs-Fe reached a photothermal effect similar to those obtained by MW and CO (~50 °C), indicating that the organic molecules did not affect light conversion into heat. Cytotoxicity studies reflected low toxicity of all the MNPs-Fe for concentrations lower than 250 µg/mL, with an improvement in the viability of the macrophages at 8 days with 50GS-MNPs-Fe. Likewise, the 50GS-MNPs-Fe sample inhibited *S. aureus*, but not wholly *E. coli*, since Gram-negative bacteria have a robust bacterial wall structure (lipopolysaccharides, peptidoglycan, and phospholipids). These results indicate that 50GS-MNPs-Fe are suitable candidates as broad-spectrum PTAs in antibacterial PTT, MRI, and magnetic hyperthermia.

## Figures and Tables

**Figure 1 ijms-24-04770-f001:**
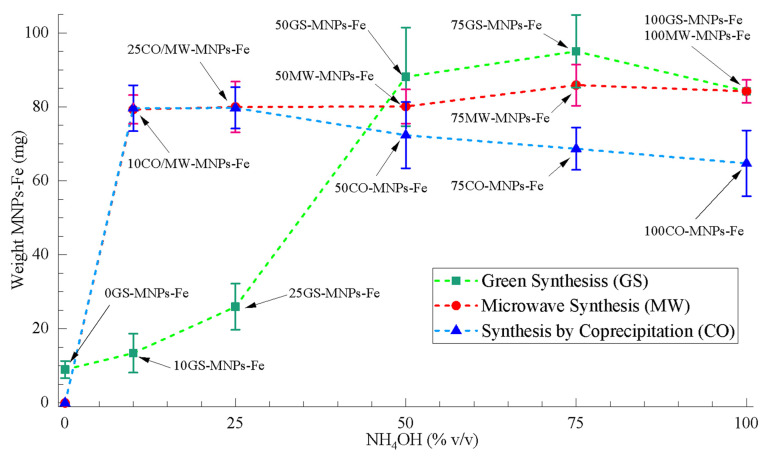
Weight curves of MNPs-Fe obtained from GS, CO, and MW with respect to the NH_4_OH (% v/v).

**Figure 2 ijms-24-04770-f002:**
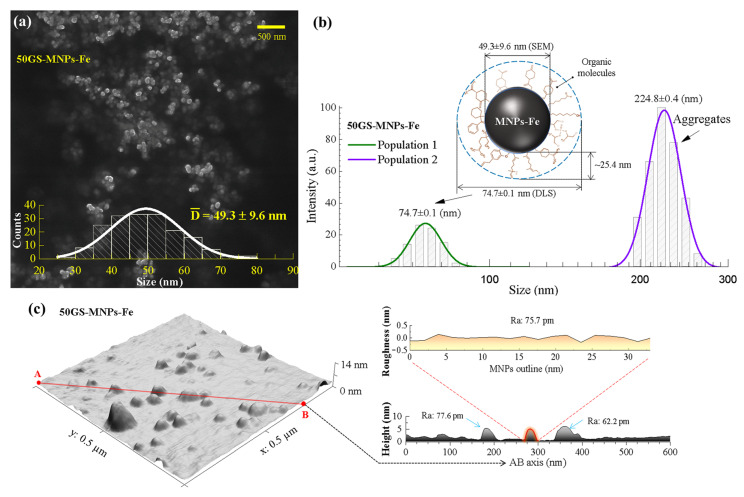
The 50GS-MNPs-Fe sample: (**a**) SEM image, (**b**) hydrodynamic size distribution obtained by DLS and (**c**) AFM image along with two-dimensional topography profiles.

**Figure 3 ijms-24-04770-f003:**
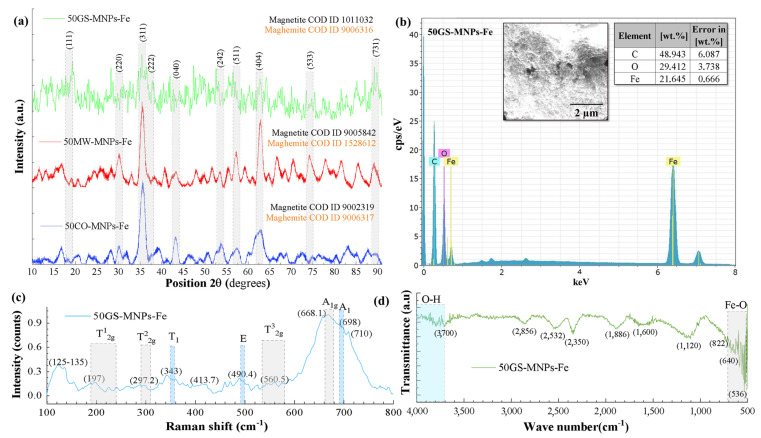
(**a**) XRD patterns of 50GS-MNPs-Fe, 50MW-MNPs-Fe, and 50CO-MNPs-Fe samples; 50GS-MNPs-Fe sample: (**b**) EDS spectrum and elemental composition (wt. %), (**c**) Raman spectrum between 100 and 800 cm^−1^ and (**d**) FT-IR spectrum.

**Figure 4 ijms-24-04770-f004:**
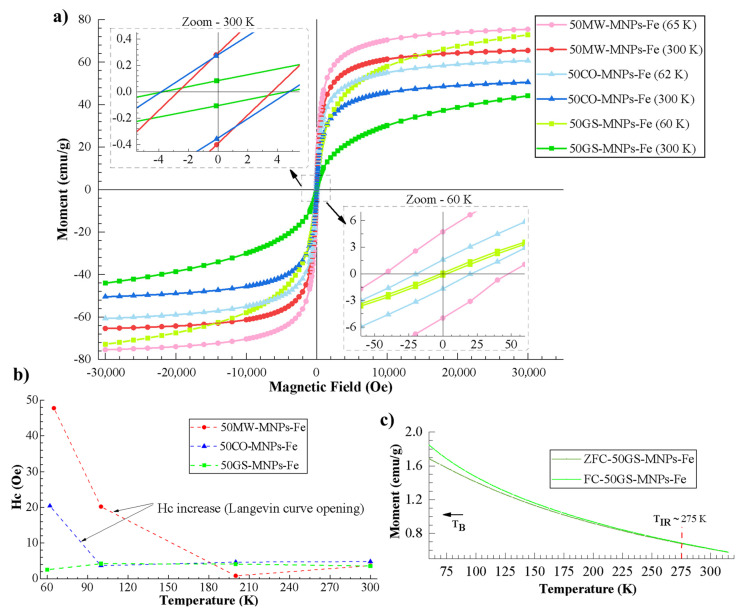
(**a**) M vs. H curves of the samples 50GS-MNPs-Fe, 50MW-MNPs-Fe, and 50CO-MNPs-Fe at different temperatures (60, 62, 65 and 300 K), (**b**) Hc vs. temperature curves of 50GS-MNPs-Fe, 50MW-MNPs-Fe, and 50CO-MNPs-Fe samples, (**c**) ZFC/FC curves of 50GS-MNPs-Fe sample.

**Figure 5 ijms-24-04770-f005:**
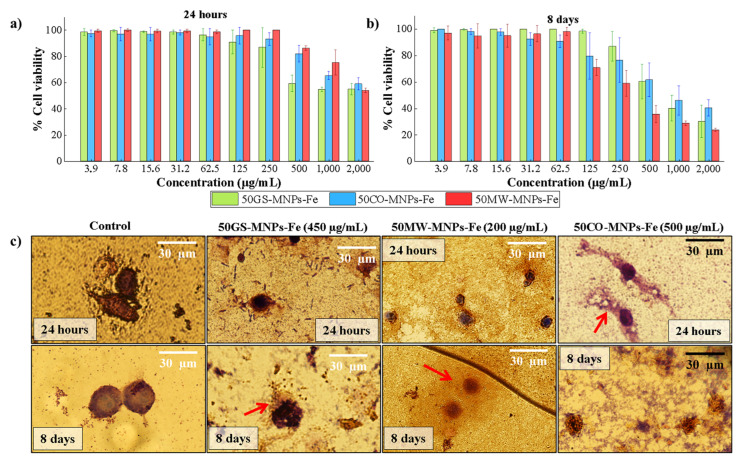
(**a,b**) Cell viability percentage in ATCC RAW 264.7 cells by Resazurin cell viability assay, using 50GS-MNPs-Fe, 50MW-MNPs-Fe, and 50CO-MNPs-Fe in 3.9–2000 µg·mL^−1^ concentrations for (**a**) 24 h and (**b**) 8 days, (**c**) Images obtained with the confocal microscope of ATCC RAW 264.7 cells incubated with 50GS-MNPs-Fe, 50MW-MNPs-Fe, and 50CO-MNPs-Fe to 450 µg·mL^−1^, 200 µg·mL^−1^, and 500 µg·mL^−1^, respectively, at 24 h and 8 days. All images are at 10× magnification. The standard errors for IC50 for 24 h are 5.992, 4.361, 3.368 µg·mL^−1^ for 50GS-MNPs-Fe, 50CO-MNPs-Fe, and 50MW-MNPs-Fe, respectively (see Appendix A), and for 48 h they are 6.855, 8.898 and 6.555 µg·mL^−1^ for 50MW-MNPs-Fe, 50CO-MNPs-Fe and 50GS-MNPs-Fe, respectively (see Appendix A).

**Figure 6 ijms-24-04770-f006:**
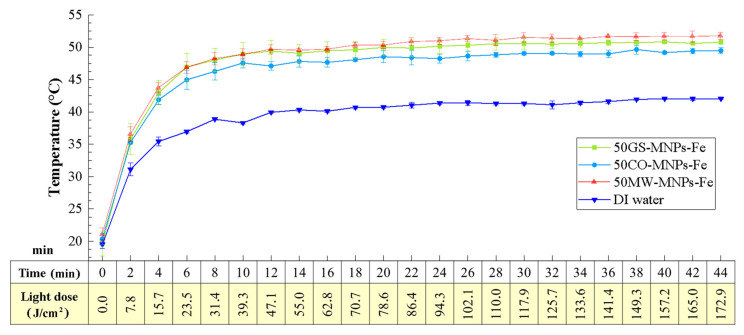
Thermal study of aqueous solutions with 50GS-MNPs-Fe, 50MW-MNPs-Fe, and 50CO-MNPs-Fe samples at concentrations of 8 mg·mL^−1^ and the control sample of DI water with red light irradiation at 630 nm and 65.5 mW·cm^−2^ from an LED source.

**Figure 7 ijms-24-04770-f007:**
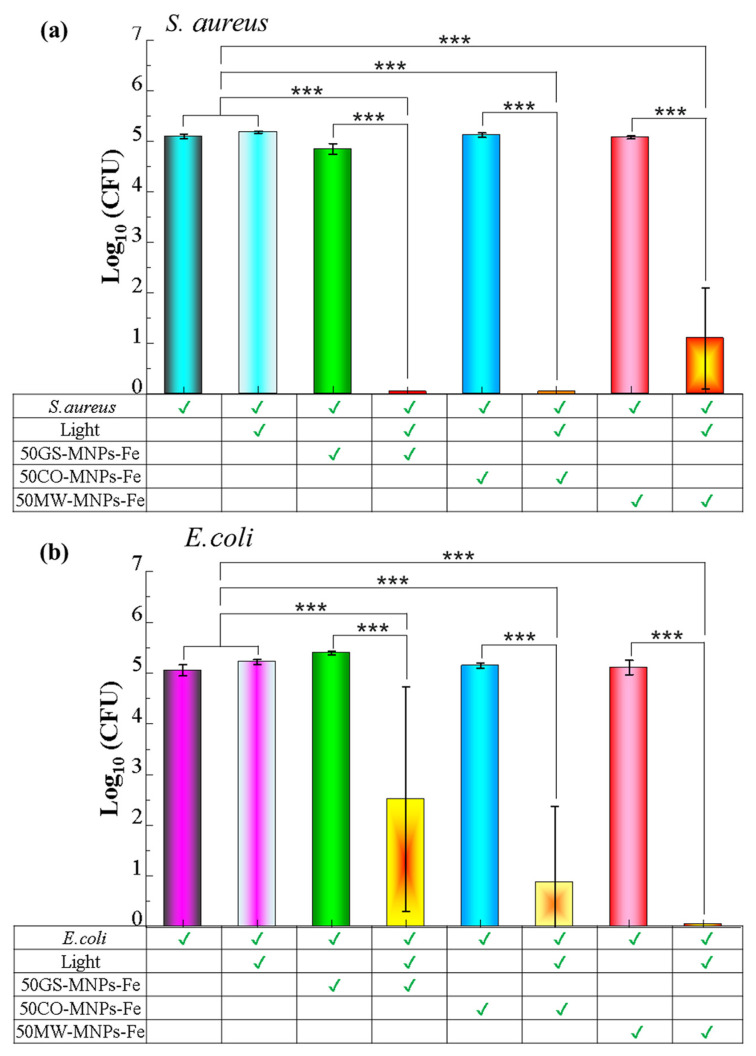
Counting of Log_10_ (CFU) in the spread plate assay to groups; [Bacteria], [Bacteria + Light], [Bacteria + MNPs-Fe (GS, MW, CO)], and [Bacteria + MNPs-Fe (GS, MW, CO) + light], in (**a**) *S. aureus* y (**b**) *E. coli*. Significant differences in means according to the Tukey test (*** *p* ≤ 0.001) based in Appendix A–c.

**Figure 8 ijms-24-04770-f008:**
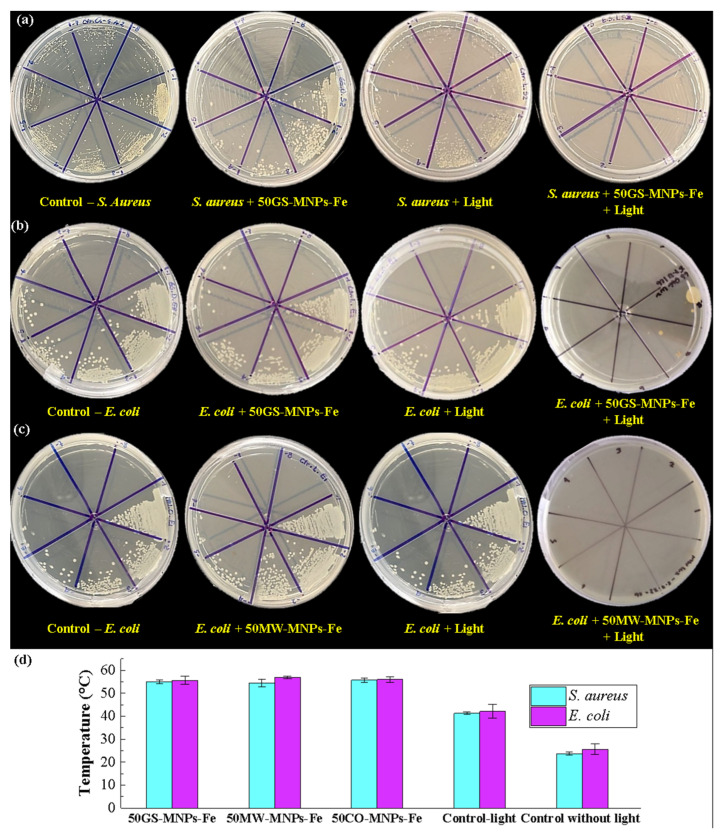
Images of the spread plate assay of the groups: (**a**) [Control-*S. aureus*], [*S. aureus* + 50GS-MNPs-Fe], [*S. aureus* + Light], [*S. aureus* + 50GS-MNPs-Fe + Light], (**b**) [Control-*E. coli*], [*E. coli* + 50GS-MNPs-Fe], [*E. coli* + Light], [*E. coli* + 50GS-MNPs-Fe + Light], (**c**) [Control-*E. coli*], [*E. coli* + 50MW-MNPs-Fe], [*E. coli* + Light], [*E. coli* + 50MW-MNPs-Fe + Light], (**d**) Temperatures reached in antibacterial PTT after irradiation with 117.9 J·cm^−2^ and control without irradiation.

**Table 1 ijms-24-04770-t001:** Mass yield and iron oxide phases of MNPs-Fe obtained by GS, MW, and CO syntheses.

Sample	Mass Yield	Phases
0GS-MNPs-Fe	1.12%	35.3% Magnetite, 64.7% Hematite
25GS-MNPs-Fe	3.71%	39.5% Magnetite, 53% Maghemite, 7.5% Hematite
50GS-MNPs-Fe	29.36%	47.0% Magnetite, 53.0% Maghemite
75GS-MNPs-Fe	18.40%	47.0% Magnetite, 53.0% Maghemite
100GS-MNPs-Fe	91.50%	6.6% Magnetite, 93.4% Maghemite
25CO-MNPs-Fe	89.08%	88.0% Magnetite, 12.0% Maghemite
50CO-MNPs-Fe	80.15%	63.5% Magnetite, 36.5% Maghemite
75CO-MNPs-Fe	74.69%	8.9% Magnetite, 91.1% Maghemite
100CO-MNPs-Fe	72.56%	98.0% Magnetite, 2.0% Maghemite
25MW-MNPs-Fe	87.73%	34.1% Magnetite, 65.9% Maghemite
50MW-MNPs-Fe	89.59%	91.2% Magnetite, 8.8% Maghemite
75MW-MNPs-Fe	93.43%	10.0% Magnetite, 90.0% Maghemite

## Data Availability

Additional results found in this work are presented in Appendix A, which will be attached to this document.

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
