# Peer review of "Bioinspired Synthesis of Magnetic Nanoparticles Based on Iron Oxides Using Orange Waste and Their Application as Photo-Activated Antibacterial Agents"

_ijms, 2023, doi:10.3390/ijms24054770_

Round 1

Reviewer 1 Report

David et. al., have submitted a paper titled (Anti-Bacterial Iron Oxide Magnetic Nanoparticles based on Orange Peel Extract). The work is interesting but there are the following major concerns that need to elaborate before publications.

1.     Authors should provide the TEM image of NPs.

2.     The topic of the paper is ambiguous and suggested to fix to make the clear sense of the use of orange peel extract.

3.     Authors should elaborate the structure property relationship of synthesized nanomaterial.

4.     Authors are suggested to discuss the antibacterial properties of some other metal based nanomaterials in the introduction section and can use the citations like (https://doi.org/10.1016/j.jsamd.2022.100417, doi:10.3762/bjnano.11.96, https://doi.org/10.1016/j.matdes.2020.109269)

5.     In Figures authors use commas “,” for numerical values it should be point “.” Like 49,3 should be 49.3. Authors should fix these in all the figures and tables.

6.     The numbers of figures are not right in captions and text so authors should fix it very carefully, for example, the first two figures have same number (Figure 1). Similarly, in text, figures are cited to number 8 but the number of figures in captions is mentioned to 7.

7.     Figure 7 (a), The labelling of bacterial culture and figure caption are not in sequence. Authors need to fix it.

8.     Authors have presented the results of bacterial culture only for S. aureus but there is the lack of culture plate results of E. coli. Authors need to provide it.

9.     In Figure 7 (b) there should be results of control without light.

10. In 2.3 cell viability. Authors claimed the cell viability after 24 h and 8 days. Why after 8 days specifically? What if results are taken after 2, 4, 6 or 10 days? Authors need to explain this point in the text and in figures accordingly.

11.  After laser irradiation there is a Fenton type reaction by iron oxide NPs that leads to reactive oxygen species. So, authors should elaborate it.

12. With reactive oxygen species there will be Photodynamic therapy (PDT) rather than Photothermal therapy (PTT). Authors need to explain it with respect to experimental data.

Author Response

Response to Reviewer 1 comments - Manuscript ID ijms-2144178

Comments from reviewer in black.

Authors replies in red.

Amendments to the text in the manuscript were made by "Track Changes". The positon of each amendment is referred to the lines using “Simple Markup”.

Comments to the Authors:

David et. al., have submitted a paper titled (Anti-Bacterial Iron Oxide Magnetic Nanoparticles based on Orange Peel Extract). The work is interesting but there are the following major concerns that need to elaborate before publications.

Response: Thank you very much for your feedback, here below are the responses for each remark.

MAJOR CONCERS

  1. Authors should provide the TEM image of NPs.

Response 1): We appreciate your suggestion and we try hard to include this requirement in our work. However, the TEM images that were acquired did not have adequate resolution to be used in the manuscript. We understand your concern about the results of the size and morphology of the NPs, however, we strongly believe that the SEM, AFM and DLS data adequate supported the discussion and elucidated conclusions.

  1. The topic of the paper is ambiguous and suggested to fix to make the clear sense of the use of orange peel extract.

Response 2): Thank you very much for your comment. To avoid ambiguities in the topic of the article, we have rewritten the article's title by " Bioinspired Synthesis Of Magnetic Nanoparticles based on Iron Oxides Using Orange Waste And Their Application As Photo-activated Antibacterial Agents". Likewise, we rewrote the Abstract and Introduction.

  1. Authors should elaborate the structure property relationship of synthesized nanomaterial.

Response 3): Thanks for this important recommendation. We have described the relationship between the NPs structure and their properties along the manuscript. Please check Section 2.2.2 in the following lines 260-266, 283-289, 362-366. Besides, complementary information was added in Table S4. Likewise, a related brief description was written in lines 376-380 and 410,411, 423.

  1. Authors are suggested to discuss the antibacterial properties of some other metal-based nanomaterials in the introduction section and can use the citations like (https://doi.org/10.1016/j.jsamd.2022.100417, doi:10.3762/bjnano.11.96, https://doi.org/10.1016/j.matdes.2020.109269)

Response 4): We have improved the Section 1 of the Introduction by adding the suggested literature about antibacterial nanomaterials as metal oxides. Please, check the 56-64 lines.

  1. In Figures authors use commas “,” for numerical values it should be point “.” Like 49,3 should be 49.3. Authors should fix these in all the figures and tables.

Response 5): Thanks for making us aware of this. We have replaced the “,” by “.”, in all the Figures as well as in the whole manuscript.

  1. The numbers of figures are not right in captions and text so authors should fix it very carefully, for example, the first two figures have same number (Figure 1). Similarly, in text, figures are cited to number 8 but the number of figures in captions is mentioned to 7.

Response 6): Thanks you for this observation. We have thoroughly reviewed and corrected these errors.

  1. Figure 7 (a), The labelling of bacterial culture and figure caption are not in sequence. Authors need to fix it.

Response 7): Thanks for the comment. We have corrected the sequence in the description of the Figure 7 (a), now Figure 8 (a) in the new version of the manuscript.

  1. Authors have presented the results of bacterial culture only for S. aureus but there is the lack of culture plate results of E. coli. Authors need to provide it.

Response 8): We have included in Figure 8b,c the cultures corresponding to E. coli for the NPs 50-GS-MNPs-Fe.

  1. In Figure 7 (b) there should be results of control without light.

Response 9): Thank you for this suggestion. We have included in Figure 7(b), now 8(d) the results of the control samples without light.

  1. In 2.3 cell viability. Authors claimed the cell viability after 24 h and 8 days. Why after 8 days specifically? What if results are taken after 2, 4, 6 or 10 days? Authors need to explain this point in the text and in figures accordingly.

Response 10): Addressing this comment, we have to indicate that typically, the cell viability assays are performed in periods of 24 and 48 hours. Nevertheless, we established an additional period of incubation using an extended time, since we wanted to corroborate the biocompatibility of the nanomaterials [1]–[9]. The period of 8 days was selected motivated in the study by Guldris et al. [10], who carried out in vivo and in vitro tests and note that from this time the results stabilized. In section 2.3, in lines 413-425, we explain this point as you suggested.

  1. After laser irradiation there is a Fenton type reaction by iron oxide NPs that leads to reactive oxygen species. So, authors should elaborate it.

Response 11): Thank you very much for this important observation. Indeed, according to the literature photothermal and photodynamic effects occur synergistically. Regarding the latter, Fe+2 and Fe+3 ions produce the Haber-Weiss / Fenton Reactions that induce ROS, increasing with the presence of photons. Unfortunately, we were unable to carry out the assay that measures ROS, since we do not have oxidizable substrates to detect singlet oxygen (1,3-Diphenylisobenzofuran). However, the contribution of ROS in our experiments was clarified in section 2.5, lines 498-503.

  1. With reactive oxygen species there will be Photodynamic therapy (PDT) rather than Photothermal therapy (PTT). Authors need to explain it with respect to experimental data.

Response 12): Thank you for your comment. According to the bibliography, PTT and PDT are produced synergistically [11]–[15]. Our results (Figure 8b and Figure 6, in the new version of the manuscript) corroborated a photothermal effect acquired by MNPs-Fe (plasmonic nanoparticles), but a photodynamic effect (increased ROS production, specifically singlet oxygen) can also be produced. This was clarified in section 2.5, lines 498-503.

References.

[1]      S. Khan et al., “Three-dimensionally microporous and highly biocompatible bacterial cellulose-gelatin composite scaffolds for tissue engineering applications,” RSC Adv., vol. 6, no. 112, pp. 110840–110849, 2016, doi: 10.1039/C6RA18847H.

[2]      J. Huang et al., “Eccentric magnetic microcapsules for orientation-specific and dual stimuli-responsive drug release,” J. Mater. Chem. B, vol. 3, no. 22, pp. 4530–4538, 2015, doi: 10.1039/c5tb00263j.

[3]      L. Xia et al., “Akermanite bioceramics promote osteogenesis, angiogenesis and suppress osteoclastogenesis for osteoporotic bone regeneration,” Sci. Rep., vol. 6, no. September 2015, pp. 1–17, 2016, doi: 10.1038/srep22005.

[4]      Y. Li, Z. Wang, and R. Liu, “Superparamagnetic α-fe2o3/fe3o4 heterogeneous nanoparticles with enhanced biocompatibility,” Nanomaterials, vol. 11, no. 4, pp. 1–14, 2021, doi: 10.3390/nano11040834.

[5]      H. Hildebrand, D. Kühnel, A. Potthoff, K. Mackenzie, A. Springer, and K. Schirmer, “Evaluating the cytotoxicity of palladium/magnetite nano-catalysts intended for wastewater treatment,” Environ. Pollut., vol. 158, no. 1, pp. 65–73, 2010, doi: 10.1016/j.envpol.2009.08.021.

[6]      M. Singh, R. K. Singh, S. K. Singh, S. K. Mahto, and N. Misra, “In vitro biocompatibility analysis of functionalized poly(vinyl chloride)/layered double hydroxide nanocomposites,” RSC Adv., vol. 8, no. 71, pp. 40611–40620, 2018, doi: 10.1039/C8RA06175K.

[7]      J. Wang et al., “The analysis of viability for mammalian cells treated at different temperatures and its application in cell shipment,” PLoS One, vol. 12, no. 4, pp. 1–16, 2017, doi: 10.1371/journal.pone.0176120.

[8]      D. G. Soares et al., “Odontogenic differentiation potential of human dental pulp cells cultured on a calcium-aluminate enriched chitosan-collagen scaffold,” Clin. Oral Investig., vol. 21, no. 9, pp. 2827–2839, 2017, doi: 10.1007/s00784-017-2085-3.

[9]      S. K. Padmanabhan et al., “Mechanical and biological properties of magnesium-and silicon-substituted hydroxyapatite scaffolds,” Materials (Basel)., vol. 14, no. 22, 2021, doi: 10.3390/ma14226942.

[10]    N. Guldris et al., “Magnetite Nanoparticles for Stem Cell Labeling with High Efficiency and Long-Term in Vivo Tracking,” Bioconjug. Chem., vol. 28, no. 2, pp. 362–370, 2017, doi: 10.1021/acs.bioconjchem.6b00522.

[11]    M. P. Romero et al., “Graphene Oxide Mediated Broad-Spectrum Antibacterial Based on Bimodal Action of Photodynamic and Photothermal Effects,” Front. Microbiol., vol. 10, no. January, pp. 1–15, 2020, doi: 10.3389/fmicb.2019.02995.

[12]    Y. Chen, Y. Gao, Y. Chen, L. Liu, A. Mo, and Q. Peng, “Nanomaterials-based photothermal therapy and its potentials in antibacterial treatment,” Journal of Controlled Release, vol. 328, no. September. Elsevier, pp. 251–262, 2020. doi: 10.1016/j.jconrel.2020.08.055.

[13]    Y. Hou et al., “ROS-responsive Ag-TiO2 hybrid nanorods for enhanced photodynamic therapy of breast cancer and antimicrobial applications,” J. Sci. Adv. Mater. Devices, vol. 7, no. 2, p. 100417, 2022, doi: 10.1016/j.jsamd.2022.100417.

[14]    F. Sozmen, M. Kucukoflaz, M. Ergul, and Z. D. S. Inan, “Nanoparticles with PDT and PTT synergistic properties working with dual NIR-light source simultaneously,” RSC Adv., vol. 11, no. 4, pp. 2383–2389, 2021, doi: 10.1039/d0ra09954f.

[15]    J. Kadkhoda, A. Tarighatnia, J. Barar, A. Aghanejad, and S. Davaran, “Recent advances and trends in nanoparticles based photothermal and photodynamic therapy,” Photodiagnosis Photodyn. Ther., vol. 37, p. 102697, Mar. 2022, doi: 10.1016/J.PDPDT.2021.102697.

Reviewer 2 Report

The present work entitled ‘Anti-bacterial iron oxide magnetic nanoparticles based on orange peel extract’ reported by David Giancarlo García Vélez is well organized with lucid language. Moreover, the findings of the work are interesting, but similar work is already online by the same authors by preprint.org (García, D.G.; Garzón-Romero, C.C.; Salazar, M.A.; Lagos, K.J.; Campana, K.O.; Debut, A.; Vizuete, K.; Rivera, M.R.; Niebieskikwiat, D.; Benítez, M.J.; Romero, M.P. Anti-Bacterial Iron Oxide Magnetic Nanoparticles based on Orange Peel Extract. Preprints 2022, 2022120436 (doi: 10.20944/preprints202212.0436.v1).) The published work is not peer-reviewed, but the preprint article must be removed from the research hub before getting acceptance from the journal (IJMS). However, the paper should be reconsidered after a minor revision before consideration.

Comments;

1.      Why author choose orange peel extract’ to synthesize iron oxide magnetic nanoparticles, although there are several works that have been published on the orange peel extract’ 

2.      Introduction should be focused on the title. Bioinspired or peel-mediated work should be cited. The introduction should be shortened. The introduction part should be revised according to the content and heading of the title.

3.      Authors are appealed to cite the following works to enhance biogenic literature in the introduction section-

Materials Science and Engineering C, 99 (2019) 783-793; Materials Today: Proceedings, 29 (2020) 720-725; ACS Omega 7 (2022), 6869−6884.

1.      The MNPs-Fe powder, drying was carried out in an oven at 40 °C for 42 532 hours. Why authors choose 40 °C for drying, and why not other temperatures? Because lower temperature might have contained hydration.

2.      Provide JCPDS no in the text with an estimate the crystallite size of all materials.

3.      Please provide sufficient control experiments (+ve and –ve) to validate the biological data collected. Also, compare all pathogen's results with standard antibiotics. How many times run the sample?

Author Response

Response to Reviewer 2 comments - Manuscript ID ijms-2144178

Comments from reviewer in black.

Authors replies in red.

Amendments to the text in the manuscript were made by "Track Changes". The positon of each amendment is referred to the lines using “Simple Markup”.

Comments and Suggestions for Authors:

The present work entitled ‘Anti-bacterial iron oxide magnetic nanoparticles based on orange peel extract’ reported by David Giancarlo García Vélez is well organized with lucid language. Moreover, the findings of the work are interesting, but similar work is already online by the same authors by preprint.org (García, D.G.; Garzón-Romero, C.C.; Salazar, M.A.; Lagos, K.J.; Campana, K.O.; Debut, A.; Vizuete, K.; Rivera, M.R.; Niebieskikwiat, D.; Benítez, M.J.; Romero, M.P. Anti-Bacterial Iron Oxide Magnetic Nanoparticles based on Orange Peel Extract. Preprints 2022, 2022120436 (doi: 10.20944/preprints 202212.0436.v1).) The published work is not peer-reviewed, but the preprint article must be removed from the research hub before getting acceptance from the journal (IJMS). However, the paper should be reconsidered after a minor revision before consideration.

Response: Thank you very much for the feedback. Regarding the published work that you mention, it actually corresponds to the Manuscript ID ijms-2144178. The preprint version of this article is available online only with the purpose to increase its visibility. It is worth mentioning that on a previous occasion we published an article that already had a preprint version, since the journal has the option of generating this document.

CONSIDERATIONS

  1. Why author choose orange peel extract’ to synthesize iron oxide magnetic nanoparticles, although there are several works that have been published on the orange peel extract’?

Response 1): Locally, there is a street trade for orange juice. This activity generates residues (among them, the peel) which are disposed of as garbage. Therefore, our work was sought the harnessing of these valuable residues by extracting organic compounds from the orange peel. The obtained extract was proposed as a potential stabilizing and functionalizing agent to synthesize magnetic nanoparticles based on iron oxides (MNPs-Fe) through a green approach. This allows the incorporation of waste from orange to the circular economy. Besides, we study the influence of the incorporation of these organic compounds on the antibacterial properties of the MNPs-Fe when they are irradiated.

  1. Introduction should be focused on the title. Bioinspired or peel-mediated work should be cited. The introduction should be shortened. The introduction part should be revised according to the content and heading of the title.

Response 2): Thank you for your valuable comment. We have reduced and improved the content of the Introduction according to the title (lines 45-139). Likewise, we rewrote the title of the article by “Bioinspired Synthesis of Magnetic Nanoparticles Based on Iron Oxides Using Orange Waste and Their Application as Photo-activated Antibacterial Agents” and also corrected the Abstract (lines 22-40).

  1. Authors are appealed to cite the following works to enhance biogenic literature in the introduction section- Materials Science and Engineering C, 99 (2019) 783-793; Materials Today: Proceedings, 29 (2020) 720-725; ACS Omega 7 (2022), 6869−6884.

Response 3): Thank you for your suggestion. We have considered the recommended articles and incorporated them in the Introduction section (lines 72-74).

  1. The MNPs-Fe powder, drying was carried out in an oven at 40 °C for 42 532 hours. Why authors choose 40 °C for drying, and why not other temperatures? Because lower temperature might have contained hydration.

Response 4): The temperature of 40 °C was selected to avoid the degradation of volatile organic compounds (VOC), such as terpenes and aldehydes, which were incorporated into the surface of the MNPs-Fe. It is recommended that these VOC, which came from the orange peel extract, are kept at 60 °C maximum. The presence of VOC on the surface of the MNPs-Fe is essential to prevent agglomeration and to improve the biocompatibility of the MNPs-Fe. Thus, we carried out a low temperature drying, but using a long period of 40 h. This condition triggered negligible remaining water in the NPs, which was barely identified by the 3700cm-1 peak in FTIR.

  1. Provide JCPDS no in the text with an estimate the crystallite size of all materials.

Response 5): Since we do not have access to JCPDS base data, we employed the Crystallography Open Database (COD) by Match! 3 software. Therefore, we ascribed the samples by identifier COD ID. We have replaced the Entry # (e.g., Magnetite 96-101-1033 COD) by COD ID (e.g., COD ID 1011032, for magnetite). Besides, the crystallite size of the samples 50GS-MNPs-Fe, 50MW-MNPS-Fe, and 50COMNPs-Fe was reported in the manuscript (lines 360-366). The corresponding information of the rest of the samples was included in the Supplementary Material in Figures S3a-S3c and S4.

  1. Please provide sufficient control experiments (+ve and –ve) to validate the biological data collected. Also, compare all pathogen's results with standard antibiotics. How many times run the sample?

Response 6): To address your comment we have elaborated the Tables S5 a-c and S6 a and b, there are detailed the experimental data of control experiments (+ve and -ve). In addition, we specified the employed runs in each assay. Please check line 618 to cell viability assays, line 675 to photothermal studies, line 650 to spread plate assay, and line 661 to the temperatures reached in the PTT. We were unable to compare our results with antibiotics because they were not planned to be performed in the present study. We could not make a comparison of the results obtained in this work with the existing literature, since no studies were found that used the same conditions as our research. Therefore, they would not be entirely comparable.

Reviewer 3 Report

1. Check the number of figures and tables.

2. The way of writing ion is not appropriate.

Author Response

Response to Reviewer 3 comments - Manuscript ID ijms-2144178

Comments from reviewer in black.

Authors replies in red.

Amendments to the text in the manuscript were made by "Track Changes".

Comments and Suggestions for Authors:

  1. Check the number of figures and tables.

Response 1: Thank you very much for your observation. We have reviewed and corrected the number and sequence of all figures and tables.

  1. The way of writing ion is not appropriate.

Response 2: Thanks for your feedback, we have appropriately write the superscript of the ions in the whole manuscript.

Round 2

Reviewer 1 Report

The revised manuscript is acceptable